# Risk Factors Associated with the Clinical Outcomes of COVID-19 and Its Variants in the Context of Cytokine Storm and Therapeutics/Vaccine Development Challenges

**DOI:** 10.3390/vaccines9080938

**Published:** 2021-08-23

**Authors:** John Hanna, Padmavathi Tipparaju, Tania Mulherkar, Edward Lin, Victoria Mischley, Ratuja Kulkarni, Aliyah Bolton, Siddappa N. Byrareddy, Pooja Jain

**Affiliations:** 1Department of Microbiology and Immunology, Drexel University College of Medicine Philadelphia, 2900 Queen Lane, Philadelphia, PA 19129, USA; jh3743@drexel.edu (J.H.); pvt24@drexel.edu (P.T.); thm35@drexel.edu (T.M.); ewl35@drexel.edu (E.L.); vmm75@drexel.edu (V.M.); ratujak@gmail.com (R.K.); arb432@drexel.edu (A.B.); 2Department of Pharmacology and Experimental Neuroscience, University of Nebraska Medical Center, Omaha, NE 68198, USA; sid.byrareddy@unmc.edu

**Keywords:** SARS-CoV-1, SARS-CoV-2, COVID-19, IL-6, cytokine storm

## Abstract

The recent appearance of SARS-CoV-2 is responsible for the ongoing coronavirus disease 2019 (COVID-19) pandemic and has brought to light the importance of understanding this highly pathogenic agent to prevent future pandemics. This virus is from the same single-stranded positive-sense RNA family, Coronaviridae, as two other epidemic-causing viruses, SARS-CoV-1 and MERS-CoV. During this pandemic, one crucial focus highlighted by WHO has been to understand the risk factors that may contribute to disease severity and predict COVID-19 outcomes. In doing so, it is imperative to understand the virology of SARS-CoV-2 and the immunological response eliciting the clinical manifestation and progression of COVID-19. In this review, we provide clinical data-based analyses of how multiple risk factors (such as sex, race, HLA genotypes, blood groups, vitamin D deficiency, obesity, smoking, and asthma) contribute to the inflammatory overactivation and cytokine storm (frequently seen in COVID-19 patients) with a focus on the IL-6 pathway. We also draw comparisons to the virulence and pathophysiology of SARS and MERS to establish parallels in immune response and discuss the potential for therapeutic approaches that may limit disease progression in patients with higher risk profiles than others. Moreover, we cover the latest information on approved or upcoming COVID-19 vaccines. This paper also provides perspective on emerging variants and associated opportunistic infections such as black molds and fungus that have added to mortality in some parts of the world, such as India. This compilation of existing COVID-19 studies and data will provide an excellent referencing tool for the research, clinical, and public health communities.

## 1. Background

Severe acute respiratory syndrome (SARS) and Middle East respiratory syndrome (MERS) are human coronavirus-associated diseases caused by the viral agents SARS-CoV-1 and MERS-CoV, respectively [1]. With the frequent genetic recombination associated with coronaviruses and increasing human-animal interface activities, the novel coronavirus SARS-CoV-2 has emerged and spread among the human population. SARS-CoV-1, MERS-CoV, and SARS-CoV-2 have all been found to have zoonotic origins and later, due to antigenic shift, expanded their host range to include humans. In 2002, SARS-CoV-1 first emerged from Chinese animal markets and was later identified as originating from bats. This pandemic affected 33 different countries with 8096 total cases, including 774 fatalities, as projected in Figure 1a. Ten years later, in 2012, MERS surfaced in the Middle East, resulting in 2494 confirmed cases and 858 deaths during that year (Figure 1b). From bats, this virus had an intermediate host of dromedary camels across the Arabian Peninsula and various African regions before finally crossing over to humans [2]. Ultimately, similar to its sister outbreak, SARS-CoV-2 was first noted from exposure to animal and seafood markets in Wuhan, China. Given the high sequence similarity between SARS-CoV-2 and the SARS-like bat CoVs, the natural host of the SARS-CoV-2 may also be from the bat lineage [3,4]. Thus, the pandemic has ensued human-to-human transmission as the primary route of worldwide spread (Figure 1c) [4,5].

SARS-CoV-1, MERS-CoV, and SARS-CoV-2 are family *Coronaviridae* and the subfamily *Coronavirinae*, all belonging to the β-coronavirus genus [1]. These single-stranded positive-sense RNA viruses contain spike glycoproteins (S) on the virus’s surface, which act as virulence factors aiding in the infection of the host cell. Mainly, the S glycoprotein consists of two subunits, S1 and S2. The former’s role is to bind the cellular receptor via the receptor-binding domain (RBD), while the latter is involved in the active process of fusion and entry into the cell. A detailed structural description of a typical coronavirus is given in Figure 2a. The general life cycle is similar among the three we discuss in this review. The virus binds via the S1 subunit receptor-binding domain to the appropriate receptor for each virus. It enters the cell through direct fusion with the cell membrane or through the endocytosis/endosomal pathway. It is important to note that coronavirus receptors are predominantly found on the alveolar membrane in the lower respiratory tract. The result of receptor binding and entry is releasing the positive-sense single-stranded RNA (+ssRNA) viral genome into the host cell cytoplasm. The +ssRNA can be directly translated by the host ribosomes. The polypeptide chains produced include replicase and an RNA replicase-transcriptase complex forms to create -ssRNA from the +ssRNA strand. These -ssRNA strands are used as templates to create more genomic +ssRNA, packaged into virions, and translated to produce late viral proteins. The late viral proteins include structural proteins S, M, and E, incorporated in virion assembly, and virions then exit the host cell through budding.

Considerable attention has been given to the nonstructural protein 1 (nsp1) of CoVs as a potential virulence factor produced early after entry into the host cell and assists with the translation of the viral genome and replication. Nsp1 is important to the β-CoV genus as it inhibits host cell proliferation, inducing cell cycle arrest in G0/G1 phase. Specifically, SARS-CoV-1 nsp1 is one of the most well-characterized and has been shown to block the expression of reporter genes under the control of constitutive promoters and the inducible interferon-beta (IFN-β) promoter through blockage of antiviral signaling pathways. To do this, SARS nsp1 tightly binds to the host 40S ribosomal subunit, thus inhibiting mRNA translation and inducing endonucleolytic RNA cleavage in the 5′-UTR of cellular mRNAs. These studies have highlighted the role of nsp1 as a potential virulence factor by attenuating the innate immune response to viral infections and exacerbating viral pathogenesis [6,7].

In terms of pathogenesis, SARS-CoV-1 infects humans through the respiratory tract via droplet transmission and binds to the angiotensin-converting enzyme 2 (ACE2) receptor via the S glycoprotein (Figure 2c) [1,8]. This ACE2 receptor is expressed on many epithelial cells in the body, primarily in the lungs, kidneys, liver, and heart. Once the virus enters through the ACE2 receptor on alveolar epithelial type II cells and replicates within these cells, it causes damage to the epithelial lining, loss of cilia, and increased alveolar macrophages that can ultimately lead to respiratory failure [1]. In contrast to the SARS-CoV viruses, dipeptidyl peptidase 4 (DPP4), also called CD26, has been established as the cellular receptor for MERS-CoV entry into pneumocytes and epithelial cells of the respiratory tract via the S glycoprotein (Figure 2c) [1,8]. Although DPP4 is expressed on many cells, including both types of alveolar cells, bronchial epithelium regardless of cilia, bronchial submucosal glands, endothelium, and alveolar leukocytes, its highest expression is in the lower respiratory tract [9]. MERS-CoV, like SARS-CoV-1, has a mean incubation period of about 5 days and is characterized by symptoms including fever, sore throat, cough, shortness of breath, and in the most severe case, respiratory failure [10]. MERS and SARS present in patients on a scale ranging from the least severe cases, including asymptomatic patients and those with mild diseases, to more severe cases, including cases of pneumonia and even acute respiratory distress syndrome (ARDS) with cytokine storms, accompanied by multiple organ system failure [11]. MERS has a higher mortality rate than SARS due to the ability of MERS-CoV to decrease the expression of genes involved in the antigen-presentation pathway, attenuating the host immune response [10].

The novel coronavirus, SARS-CoV-2, similar to SARS-CoV-1, is transmitted via droplets and enters the cell via the ACE2 receptor to infect the upper and lower respiratory tract (Figure 2c). The basic reproduction ratio of infection, the R0 value, of SARS-CoV-2 is 3.28 compared to 2.2 in SARS-CoV-1 and 0.69 in MERS. This indicates that SARS-CoV-2 has an increased transmission rate compared to SARS-CoV-1, explaining the rapid global increase in coronavirus disease 19 (COVID-19) cases [4,5]. Although this virus transmits faster between individuals, the symptoms are milder than those associated with SARS and MERS, and the mortality rate among high-risk groups is also significantly lower (3.4%) than that of SARS (9.6%) and MERS (35%) [1]. The clinical and radiological manifestations of patients with confirmed COVID-19 are similar to those infected with SARS in 2003. Still, the two viral agents can be distinguished from each other based on genomic sequencing of their respective RNA-dependent RNA polymerase (RdRp) [1,8]. Another difference was illustrated in the different RBD residues in the S glycoprotein of both viruses, with 10 to 20 times increased affinity for the ACE2 receptor in the SARS-CoV-2 glycoprotein as compared to SARS-CoV-1 [1]. After a 5- to 14-day incubation period, COVID-19 patients typically manifest features of fever, dry cough, dyspnea, myalgia, and fatigue. Patients with severe cases developed ARDS along with lymphopenia and hypoalbuminemia. Other documented symptoms include headache, hemoptysis (coughing of blood), and diarrhea. Analysis of the cases of SARS-CoV-2 revealed many unique characteristics of the virus, such as the transmission of the disease via asymptomatic carriers. The most notable characteristic is the shedding of the virions starting earlier than clinical manifestations, allowing transmission during the incubation period and increasing this virus’ chances of causing a pandemic [5].

In this clinical data-based account, we covered aspects of host-pathogen interactions between the immune system and coronaviruses. We further analyzed the role of sex and race, HLA allele impact, and ABO blood group distribution and how they may impact susceptibility to these ssRNA viruses. Additionally, clinical conditions and vulnerabilities such as vitamin D deficiency, nicotine usage, asthma, and obesity are analyzed for their impact on SARS, MERS, and COVID-19 outcomes to further understand why clinical presentations vary between populations affected by these diseases.

## 2. Host-Pathogen Interaction

The innate immune system comprises a number of defense mechanisms that are in place to provide protection against foreign pathogens and eliminate damaged host cells. Its components can recognize repeating patterns of molecular structure that are common to certain classes of pathogens. Not expressed by host cells, these structures called pathogen-associated molecular patterns (PAMPs) are recognized by innate immune system receptors called pattern recognition receptors (PRRs) [13]. There is a multitude of different PRRs, each with its degree of specificity for certain PAMPs. While Toll-like receptors (TLRs) can recognize a protein, lipid, and nucleic acid PAMPs, carbohydrate residues are recognized by lectins, such as DC-SIGN (dendritic cell-specific ICAM-3 grabbing non-integrin) or its tissue-resident cousin, langerin, present on Langerhans cells [13,14]. Figure 3 provides the outcome and mechanistic of coronaviruses’ interactions with these various immune receptors. PAMP-mediated activation of PRRs upregulates transcription of genes encoding proinflammatory cytokines/chemokines, type I interferons (IFNs), and antimicrobial proteins, all involved in inflammatory responses [15]. Type I IFNs, such as IFN-α and IFN-β, play a critical role in initiating antiviral responses, as they influence protein synthesis apoptosis in infected cells. Type I IFNs also play a key role in adaptive immunity by enhancing dendritic cell (DC) maturation, natural killer (NK) cell cytotoxicity, and virus-specific cytotoxic T lymphocyte differentiation [15].

In mammals, a distinct class of PRRs critical in detecting pathogens and initiating the host immune response is known as Toll-like receptors (TLRs) [13]. The TLR family comprises 10 members (TLR1-TLR10) in humans and can be expressed in both innate immune cells such as macrophages and DCs in addition to non-immune cells such as epithelial cells and fibroblasts. These TLRs are generally divided into two groups based on their localization: either on the cell surface (TLR1–6 and TLR10) or in intracellular (endosomal) compartments (TLR3, TLR7–9). The difference between these two subfamilies depends on the components they recognize. Remarkably, cell surface TLRs recognize foreign membrane components, including lipids, lipoproteins, and proteins. In contrast, intracellular TLRs recognize foreign nucleic acids derived from microorganisms and endogenous nucleic acids often found in autoimmune disease conditions [16]. Of the four endosomal members, TLR7 and TLR8 are specific to detecting viral ssRNA and therefore initiating immune responses to these viruses [17,18]. Upon the engagement of TLR7/8 with oligonucleotides containing guanosine and uridine-rich sequences of ssRNA viruses such as SARS-CoV-1, MERS-CoV, and SARS-CoV-2 in endosomes, these antiviral sensors initiate the myeloid differentiation primary response 88 (MyD88) pathway. This pathway, in turn, induces mRNA-transcription of many proinflammatory cytokines via the nuclear factor-kappa B (NF-κB), a chief transcription responsible for regulating gene expression of innate and adaptive immune response sequences [17,19].

Another family of carbohydrate-recognizing receptors called lectins also plays an instrumental role in detecting pathogens and activating the immune response. Of these, DC-SIGN is a potent PRR expressed on DCs, macrophages, and various other DC-SIGN expressing cells and has been identified to bind to a host of viral, bacterial, and fungal pathogens [20,21]. DC-SIGN, found mainly on the surfaces of DCs, binds strongly to glycoproteins expressed by pathogens and ICAM-3 on the surfaces of T cells [14]. A type-II transmembrane protein comprises a cytoplasmic, transmembrane, repeat region, and a C-type lectin or the carbohydrate-recognition domain. Despite its role in recognizing and signaling for the endocytosis and breakdown of pathogens, the resultant of binding is not always straightforward. Numerous studies have shown that DC-SIGN can act as a facilitator of infection [22,23,24]. Studies on DC-SIGN have demonstrated that interactions of enveloped pathogens, including viruses, with DC-SIGN, leads to immune escape in non-lysosomal compartments, facilitating safe transmission to infection sites in a trojan horse manner [20,22,23,24,25]. Concurrently, inhibition of DC-SIGN by dextran and peptide triazoles results in the inhibition of HIV-1 infection, suggesting that it, too, uses DC-SIGN as an avenue of infection [26]. More relevant to current epidemiology, it has been shown that interactions between the CoV-1 S protein and DC-SIGN mediate the transfer of viral particles to target cells; the latest studies support the notion that DC-SIGN acts as an alternate receptor for the related CoV-2 virus [27,28].

Among the most important cytokines produced in response to TLR7/8 activation is interleukin-6 (IL-6), a cytokine of pleiotropic activity [13]. Figure 4 explains the classical and trans-signaling of IL-6. Soon after the stimulation of TLRs, monocytes and macrophages produce IL-6 to modulate host defense through various mechanisms. These mechanisms include (1) antibody production by activated B cells; (2) acute-phase protein production in the liver such as C-reactive protein (CRP), fibrinogen, serum amyloid A, and hepcidin; (3) megakaryocyte maturation into platelets and hematopoietic stem cell activation in bone marrow, promoting osteoclast differentiation and keratinocyte and mesangial cell proliferation [29]. These mechanisms persist until the pathological agent is removed and regulatory systems normalize serum IL-6 and CRP levels, terminating the IL-6-mediated signal transduction cascade [30]. However, if IL-6 levels remain elevated, chronic autoimmune diseases and inflammatory diseases such as respiratory failure can develop, as seen with SARS patients.

The most severe COVID-19 patient cases reported significantly increased levels of proinflammatory cytokines causing cytokine storm, which is a condition caused by overstimulation of the immune system. Several studies have pointed to a strong correlation between elevated IL-6 (prevalent in cytokine storm) levels in the serum and the development of respiratory failure in COVID-19 patients [12]. Following viral replication and disengagement of lung epithelium, the virus is captured by macrophages and DCs. It induces a high production of IL-6, which in turn activates the adaptive immune response and induces T-cell infiltration into the alveoli. Cytotoxic T cells recognize the infected cells and initiate their destruction in an attempt to eliminate the virus. IL-6 also induces the recruitment and stimulation of fibroblasts in the alveoli and the successive deposition of the extracellular matrix, which leads to respiratory failure and lung fibrosis [31]. Another mechanism that exacerbates immune response is increased blood vessel permeability, enabling invasion of effector cells to produce more inflammatory cytokines. This high blood vessel permeability also allows SARS-CoV-2 to spread within the body to infect distant ACE2-expressing tissues (kidney, intestine, pancreas), leading to a burst of inflammatory mediators at these sites [12].

An additional explanation for the overproduction of IL-6 and the induction of a cytokine storm in SARS-CoV-2 patients is related to the ACE2 receptor entry site. The entry of the virus through ACE2 reduces receptor availability, and in turn, leads to the buildup of angiotensin (Ang) II [32]. Ang II itself significantly increases IL-6 expression in a concentration-dependent manner. The accumulation of Ang II and its binding to the Ang I receptor (AT1R) results in downstream activation of NADPH oxidase and production of oxidative stress, which directly stimulates NF-κB-mediated transcription of IL-6. Once IL-6 is produced, it directly promotes AT1R expression and sensitization of the vascular wall to angiotensin IL-dependent signaling mechanisms, generating a positive feedback loop that could be a primary driver of massive IL-6 and cytokine production in these patients [12].

## 3. Risk Factors

While the biochemical pathway of the immune response against viruses can explain the host and pathogen interaction, there are many factors, both genetic and clinical, that may change the susceptibility and thus the outcome of these viral diseases. These include sex and race, genetic factors, and pre-existing conditions and vulnerabilities discussed in detail below.

### 3.1. Sex and Race

Differences in COVID-19 incidence and infection outcomes have been observed between genders and between races [33,34,35,36]. Understanding the potential mechanisms behind the differentials in infection and outcome is vital for discovering possible avenues of treatment. Some of these differences may be at the level of genetic differences, such as the expression of TLRs.

RNA viruses use endosomal TLR members such as TLR7 and TLR8 to elucidate the appropriate immune response against the pathogen [17,18]. Interestingly, the lower mortality rate associated with CoV-2 can be directly linked to the immune response initiated by the TLR7 receptor. A study performed by Moren-Eutimio et al. evaluated ssRNA fragments from the genomes of SARS-CoV-1, MERS-CoV, and SARS-CoV-2 that bind to TLR7 recognition sites. Interestingly, the CoV-2 genome contains more fragments recognized by TLR7 than the other two viruses, suggesting that SARS-CoV-2 is essential in creating a more robust TLR7-induced immune response [18]. Not only is there a discrepancy in mortality rates due to the differential recognition of the three viruses by TLR7, one can go further and evaluate the varying susceptibilities to the three viruses based on the relationship between TLR7 expression and sex. The WHO has reported that 63% of deaths related to COVID-19 have been among male patients. In New York City, the mortality rate for male patients is almost twice the rate of females.

Similarly, illness outcomes were far worse for males than females during the previous two coronavirus outbreaks. Males with SARS were found to have a significantly higher fatality rate than females did, with a relative risk of 1.62 [33]. Comparative epidemiology of MERS in Saudi Arabia and South Korea also revealed that males were at a higher risk of dying than females [34]. These findings could partially be explained by the fact that TLR7, which is transcribed on both X chromosomes in a large proportion of the plasmacytoid DCs, B cells, and monocytes, escapes the X-chromosome inactivation. Normally, genes found on the X chromosome undergo silencing to account for proper gene dosage between males and females. A study performed by Souyris et al. [37] demonstrated that with TLR7 escaping gene silencing, biallelic B lymphocytes in females displayed higher TLR7 transcription than monoallelic cells in males, thus corresponding to greater protein expression in females than in male leukocyte populations. These higher levels of TLR7 provide a selective advantage at critical TLR-dependent checkpoints of effector B cells for normal women and Klinefelter males compared to individuals with one X chromosome.

Recent research has revealed the importance of race in the contributable factors for systemic inflammation. The role of the race was studied through an observational study by White SR et al. on asthmatic patients with high circulating levels of IL-6. Patients of African American ancestry were found to have higher IL-6 concentrations than those with European American ancestry [33]. This study supported a previous cross-sectional observational study performed in 2016, which analyzed biomarkers such as C-reactive protein, IL-6, fibrinogen, and E-selectin among various racial demographics. Here, it was found that African Americans had greater concentrations of all four biomarkers of inflammation when compared with their white counterparts [38]. The impact of race on systemic inflammation can further be seen through how disproportionately COVID-19 has affected African Americans, LatinX, and Native American communities in the United States [35]. As of June 2020, the CDC reported that African Americans comprised 21.98% and LatinX comprised 33.8% of COVID-19 cases in the U.S. despite these racial and ethnic groups only comprising 13% and 18% of the total U.S. population, respectively.

Additionally, the mortality rate for COVID-19 among African Americans is two-fold more significant than that of Whites. Many reasons contribute to this strikingly apparent issue. The disproportionate burden of chronic inflammatory medical conditions such as diabetes, hypertension, obesity, and coronary artery disease is higher among these racial and ethnic groups. It has been found to increase patient risk for developing severe manifestations of COVID-19 and mortality. Furthermore, these chronic medical conditions are compounded by more inadequate access to healthcare often found among certain racial and ethnic minority groups [35,39].

The role of race has also been analyzed among other factors that are strongly associated with systemic inflammatory cytokine dysregulation, such as psychological stress and psychosocial trauma. These conditions are directly related to increased levels of systemic proinflammatory markers such as TNF-alpha, IL-6, CRP, and resistin [40], which exacerbate SARS-CoV-2 outcomes. Qualitative reports and meta-analyses have repeatedly indicated higher levels of psychological stress among African American populations compared to Whites. This psychological stress can be due to multiple reasons, including mental health conditions, socioeconomic inequalities, and occupational hazards. High levels of psychological stress can lead to decreased immunity and chronic inflammation, ultimately putting African Americans at higher risks for developing severe COVID-19 symptoms [41].

Furthermore, chronic infections can also contribute to an increased risk of COVID-19 disease. For instance, tuberculosis (TB), which is most prevalent among U.S. racial minorities, contributes to chronic lung changes such as cavitation, fibrosis, bronchiectasis, and impairment in lung function, which could all lead to impaired pulmonary immunity and increased risk for respiratory infections. Furthermore, TB, HIV, and hepatitis are all associated with a chronic elevation in proinflammatory cytokines, which increases the risk for respiratory failure in COVID-19 patients [41].

The spread of SARS-CoV-2 from contact with symptomatic and asymptomatic individuals can explain the increased transmission and mortality among African Americans. This population disproportionately endure unfavorable environmental neighborhood factors such as crowded housing within dense population areas and limited green spaces, recreational facilities, and safe transportation systems. African Americans are more likely to use public transportation than their white counterparts, resulting in a higher risk of exposure to SARS-CoV-2 [42]. The African American population maintains a more significant disease burden with greater poverty, limited health care access, higher rates of jobs in service industries with less lifestyle flexibility, and apparent food insecurity [42,43].

### 3.2. Genetic Factors

Genetic factors are likely to extend beyond the scope of influencing the infection and outcomes of different races and genders. There is significant evidence that HLA polymorphisms and blood type influence the susceptibility of individuals to infection and their outcomes [44,45,46,47,48].

#### 3.2.1. HLA Polymorphisms and Disease Manifestations

HLA genes have been implicated in a wide variety of diseases. The HLA gene is located on chromosome 6 on band 6p21.3 and encodes for the major histocompatibility complex, MHC, crucial in antigen presentation to T cells [49]. The HLA locus can be broken down into two classes. The class 1 HLA genes include HLA-A, HLA-B, and HLA-C, while the class 2 genes include: HLA-DPB1, HLA-DQA1, HLA-DQB1, HLA-DRA, and HLA-DRB1 [50]. These genes are highly polymorphic, and over 9000 alleles have been reported. HLA polymorphisms have been associated with a change in clinical outcome in a wide variety of immune-related conditions, specifically in respiratory infections; specific allelic polymorphisms have been associated with a change in risk: pulmonary tuberculosis, pulmonary macrophage *Mycobacterium avium* complex infection, diffuse Pancobronchiolitis, Sarcoidosis, MERS-CoV, SARS-CoV-1, and SARS-CoV-2 [44,45,46]. Table 1 summarizes the association of HLA alleles with SARS viruses.

In both SARS-CoV-1 and SARS-CoV-2, HLA-B46:01 has been associated with lower peptide presentation, resulting in a higher risk for the development of disease. This has been shown in three separate studies on different settings, including Taiwanese patients, a world population analysis, and an in silico analysis. HLA-B46:01 was derived from a mini conversion between HLA-B*15:01 and HLA-C*1:02 in Southeast Asia, and now is the most common HLA-B allele [51]. Specifically in China, the Southwest Dai population has an allele frequency of 0.2540 [52]. Due to the unique recombination, which resulted in HLA-B46:01, this allele possesses both the C1 epitope and a killer-immunoglobulin-like receptor ligand. Compared to both HLA-B*15:01 and HLA-C1:02, HLA-B46:01 has a higher affinity for the NK receptor, KIR2DL3. However, HLA-B46:01 has a smaller peptidome than HLA-B15:01, which may explain the susceptibility to the SARS-CoV-2 virus. Although HLA-B46:01 has a limited range of peptidome, it is protective against some infections, such as TB, as HLA-B27 has a high degree of affinity for recognized peptides. Although protective in instances such as TB, HLA-B27 is associated with the development of autoimmune conditions due to a lack of ability to limit self-reactivity [51]. For example, in a meta-analysis of Asian patients with Graves’ disease, there was an association with the HLA-B46:01 allele and the development of Graves’ disease [52]. The combination of both low recognition of peptides and a high rate of autoreactivity might explain why HLA-B46:01 was associated with the severity of SARS-CoV-1 [53]. Future research should be performed to investigate if HLA-B46:01 is related to the development of either a cytokine storm or increased severity of patients or both in patients with SARS-CoV-2.

On the other hand, HLA-DRB1*03:01 correlated with higher peptide presentation in SARS-CoV-1, which resulted in decreased susceptibility to the virus [54]. Interestingly, HLA-DRB1*03:01 has also been shown to be associated with reduced body mass index (BMI), higher prevalence in T1D females when compared to males, and increased prevalence in those of European descent. All of the populations mentioned above have been associated with less severe presentations of SARS-CoV-2 (47–51). While HLA-DRB1*03:01 was a strong presenter of SARS-CoV-1 peptides, HLA-DRB1*03:02 was a weak presenter of SARS-CoV-2 [55]. This is interesting because HLA-DRB1*03:01 is a European-derived allele, while HLA-DRB1*03:02 is an African-derived allele derived from HLA-DRB1*03 [56]. This should be further investigated to see if the difference in the prevalence of HLA-DRB1*03:01, and HLA-DRB1*03:02 could help explain the difference seen among races [57]. In addition, it was found, in silico, that the Sub-Saharan African population had the highest frequency of weak binding HLA class 2 molecules compared to other populations [55]. A paper by Barquera et al. [55] analyzed peptide binding of 438 different HLA proteins and found that class 2 MHC molecules were bound to SARS-CoV-2 peptides with a regular strength at a rate of 6%–21%, significantly higher than the class 1 MHC molecules with a rate of 2%. Class 2 MHC molecules also had a higher proportion of molecules bound weakly when compared to class 1 (39%–40% compared to 8%). This leads to the question of there is a difference in CD4^+^ or CD8^+^ T-cell expression in patients who have CoV-2. It has been consistently shown that the ratio of CD4:CD8 ratio remains unchanged throughout infection [58]. However, there was an overall decrease in CD4^+^ and CD8^+^ T cells in patients with severe cases of SARS-CoV-2 or SARS-CoV-1 compared to less severe patients [59,60]. A recent study found that decreased CD8^+^ cells correlated with poor clinical outcomes in patients with SARS-CoV-2 and were negatively correlated with inflammatory markers: ESR, CRP, and IL-6. In contrast, CD4^+^ T cells were only negatively correlated with ESR [45]. While these results indicate increased importance of CD8^+^ T cells in SARS-CoV-2, in SARS-CoV-1, 100% of patients had reduced CD4^+^ T cells, and 87% experienced a decrease in CD8^+^ T cells, with 85% experiencing lymphopenia [45]. In contrast, only 34% of patients with MERS-CoV experienced lymphopenia [61].

#### 3.2.2. HLA-DR Expression and Disease Manifestations

HLA-DR downregulation has been associated with increased mortality after septic shock, indicating immune failure [62]. A paper by Giamarellos-Bourboulis et al. [63] demonstrated that among patients with COVID-19, there was a significant downregulation of HLA-DR expression on CD14+ monocytes in both COVID-19 patients with severe immune dysregulation and with macrophage activation syndrome (MAS) when compared to both patients with intermediate severity and healthy volunteers, demonstrating a difference between the severe population and the intermediate population. Furthermore, there was an absence of IFNχ in all patients and no significant difference in TNF-α levels between patients with varying severity of COVID-19. However, there were significantly higher levels of IL-6 and CRP in patients with immune dysregulation and MAS compared to patients with intermediate COVID-19. Changes in HLA-DR expression have been linked to the downstream effects of IL-6 expression. Increased IL-6 expression, as seen in the cytokine storm brought about by many respiratory diseases, results in downregulated HLA-DR expression on human monocyte-derived DCs through the IL-6 mediated STAT3 pathway [59]. Proteins involved in the HLA-DR downregulation process include cyclooxygenase 2, lysosome protease, and arginase [59]. The downregulation of HLA-DR on human DCs hinders the activation of antigen-specific CD4^+^ T cells to present cancer-related antigens [60]. HLA-DR expression was shown to be partially restored using anti-IL-6 therapy, tocilizumab, which is in accordance with data that have demonstrated that IL-6 affects HLA-DR expression, ultimately attenuating the CD4^+^ T-cell response [63]. A functional adaptive immune system is vital in SARS-CoV-2 immunity as 100% of patients who recovered from the SARS-CoV-2 virus had specific CD4^+^ T cells [64]. Specifically, the absence mentioned above of IFN-Ɣ and the presence of IL-6 indicates a lack of Th1 response, implicating a Th2 as the significant response mechanism [63].

#### 3.2.3. Blood Type

Early during the COVID-19 pandemic, several papers proposed an association between blood type and the severity of SARS-CoV-2 infection. Overall, blood types A and AB appear to increase susceptibility to COVID-19 infection, while type O blood appears to protect against COVID-19 infection. First, type A was found to be associated with higher susceptibility to COVID-19, while type O was found to be a protective allele [65]. A preprint article in April 2020 correspondingly found a higher incidence of infection within blood groups A and AB and a lower incidence in blood type O [66]. Later (June 2020), a large multi-institutional study looked at over 7000 patients who received COVID-19 testing. The data were per the study mentioned above that blood type AB and were associated with higher odds of testing positive for COVID-19 while O had lower odds of testing positive for COVID-19. A positive association was found between the B blood group and testing positive, while no such association was determined between blood type A and rates of testing positive. Further research, however, did not find any significant correlation between blood type and inflammatory markers, clinical outcome, risk of intubation, or death [67]. These data follow a similar pattern in SARS-CoV-1 wherein O blood type was correlated with less susceptibility to SARS-CoV-1 [47]. Various cells outside of red blood cells express ABO antigens such as enterocytes, pneumocytes, and cells in the distal tubule of the kidney, all cells that are infected by the SARS viruses. It has been shown that anti-histo-blood group antibodies can neutralize HIV in vitro, and a similar mechanism could occur in the SARS viruses [68]. Blood group O has both anti-A and anti-B antibodies. These antibodies have been shown to bind to glycan-binding sites, which could spatially interfere with the binding of SARS to the ACE2 receptor, decreasing viral entry [69]. The reduced viral entry to the cell would agree with the recent SARS-CoV-2 data suggesting blood type O is associated with reduced susceptibility but not reduced severity of the disease.

### 3.3. Pre-Existing Conditions and Vulnerabilities

It has been evident that the global COVID-19 pandemic can overwhelm healthcare systems and may have varying impacts on different populations. As such, it is critical to recognize vulnerable populations with higher risk factors [70]. Pre-existing conditions such as vitamin D deficiency, obesity, and asthma have been associated with more severe symptoms in individuals with respiratory illnesses. Activities such as smoking also lead to an increased risk for severe respiratory disease [71]. Therefore, it is important to explore these factors in COVID-19 patients and evaluate their impact on disease progression.

#### 3.3.1. Vitamin D Deficiency

Fat-soluble vitamin D is obtained by humans either through diet or produced endogenously when UV rays from sunlight trigger synthesis in the skin [72]. The vitamin D receptor is expressed on immune cells, including B cells, T cells, and antigen-presenting cells, allowing these immune cells to synthesize the active form of vitamin D [73]. Vitamin D can reduce infections through the induction of cathelicidins and defensins, which lower viral replication rates and reduce the elevation of proinflammatory cytokines [74]. A deficiency in vitamin D can lead to increased autoimmunity and susceptibility to infection. The European Calcified Tissue Society Working Group has defined severe vitamin D deficiency as having serum 25(OH)D levels of lower than 30 nmol/L [75]. Several factors can lead to deficiency, such as the inability to sufficiently synthesize vitamin D [72]. Although COVID-19 can infect all age groups, most severe cases are typically seen in older populations with underlying conditions such as cardiovascular and pulmonary disease [75]. These same older populations are also most susceptible to severe vitamin D deficiency, which may further exacerbate poor immune outcomes in these patients. It is also worth noting that geographical location may also affect vitamin D synthesis: for example, the aging population in Southern Europe has the most deficient vitamin D levels given that they typically spend less time outdoors, and skin pigmentation decreases vitamin D synthesis [75]. Thus, an additional risk factor in poor SARS-CoV-2 outcomes may relate to geographical variation in relation to vitamin D deficiency.

Vitamin D has also been associated with hypertension and can impact the renin-angiotensin system. A randomized control study performed by McMullan et al. demonstrated that lower circulating 25-hydroxyvitamin D (25(OH)D) is associated with increased renin-angiotensin system (RAS) activity and thus increased blood pressure in humans [76]. Given that hypertension is a crucial risk factor for COVID-19, this further suggests that vitamin D deficiency creates a critical risk factor for severe COVID-19 outcomes. Multiple cross-sectional analyses support these findings by showing that vitamin D deficiency was a common risk factor among two countries with the highest COVID-19 mortality rates, Italy and Spain [77]. Thus, several studies have encouraged higher vitamin D3 doses as part of the treatment process for those infected by COVID-19 [74].

There have also been links between vitamin D and HLA-DR expression. In the past, vitamin D deficiency has been associated with a poorer prognosis in autoimmune conditions such as rheumatoid arthritis, lupus, and MS and an increased susceptibility toward infectious diseases such as respiratory tract infections and tuberculosis [78,79]. Recently, vitamin D has been postulated to increase susceptibility to SARS-CoV-2 due to both the outbreak occurring in the winter and the older population infected, both factors associated with vitamin D deficiency [74]. Research has shown that calcitriol, a vitamin D derivative, upregulated CD14 expression and downregulated HLA-DR expression [80]. Furthermore, vitamin D inhibited an increase in HLA-DR expression while limiting dendritic cell maturation [81]. Vitamin D may be necessary for preventing the overactivation of the immune system seen in macrophage activation syndrome or the cytokine storm experienced by patients with severe COVID-19. Moreover, the other mechanisms of vitamin D’s regulation on the immune system, such as downregulating the ACE2 receptor, could outweigh the harm done by downregulating HLA-DR and preventing DC maturation [82].

Deficiency in vitamin D has also been linked to an increase in IL-6, which is known to peak at the height of respiratory failure in COVID-19 cases. Vitamin D inhibits monocyte production of inflammatory cytokines, including IL-6, allowing healthy regulation. Therefore, vitamin D deficiency provides a risk factor for elevated IL-6 levels, and in turn, could be used as an indicator of potential severe respiratory symptoms in COVID-19 patients [83]. Previous CoV pandemics such as SARS and MERS also show evidence of vitamin D deficiency-related complications. Vitamin D deficiency can contribute to complications such as cytokine storms, which were seen in many SARS and MERS cases. Due to observations of low vitamin D levels and high IL-6 expression in SARS, MERS, and COVID-19 patients, vitamin D supplementation may be considered as a useful measure to prevent dangerous symptoms from developing in viral respiratory infections [74].

#### 3.3.2. Obesity

An analysis of the relationship between obesity and COVID-19 was executed in a French center. They studied clinical characteristics such as BMI and the requirement for invasive mechanical ventilation in 124 patients. The high frequency of obesity in the admitted patients supported the conclusion that COVID-19 symptoms increased in severity with BMI, signifying obesity as a risk factor for the disease [84]. Additionally, obesity is a critical risk factor to consider with COVID-19 due to its relationship with vitamin D deficiency. A study performed by Wortsman et al. assessed whether obesity alters cutaneous production of vitamin D3 or the intestinal absorption of vitamin D2. The subjects in the obese category had significantly lower vitamin D concentrations but higher parathyroid hormone concentrations [85]. This study concluded that this deficiency is likely due to decreased bioavailability of vitamin D3 from cutaneous and dietary sources due to the accumulation of vitamin D3 in fat compartments. The postulation was that obese individuals may have a more incredible difficulty converting vitamin D, resulting in a deficiency that may make them more vulnerable to COVID-19. The impact of obesity on vitamin D production can be seen as a risk factor in the other CoVs, as obese patients with MERS were thought to be more likely to risk severe disease or death [86]. Obesity is directly associated with systemic inflammation through its tie with the cytokine IL-6. In addition to macrophages that are fixed within the fat tissue, adipocytes are known to be a major source of IL-6 found in the blood. The adipokine leptin is also known to stimulate the release of IL-6 from leukocytes and macrophages. In obese white adipose tissue, proinflammatory M1 macrophages form crown-like structures surrounding the dead adipocytes, contributing to the obesity-induced low-grade inflammation and production of inflammatory cytokines such as IL-6. With a clear association between obesity and IL-6 levels evident, it is probable that genetic factors influencing weight gain would also have a corresponding effect on IL-6 and may become a risk factor for COVID-19 patients [87].

#### 3.3.3. Smoking, Nicotine, and Asthma

Smoking is known to increase susceptibility to respiratory infection [71]. Cigarette smoke contains numerous toxic chemicals, and the action of smoking is connected to several well-established morbidities. Many smokers suffer from comorbidities such as cardiovascular disease, which is a severe risk factor for COVID-19. The World Health Organization has identified smokers as a vulnerable and susceptible population to COVID-19. Furthermore, significantly higher ACE2 gene expression has been found in smoker samples versus non-smoker samples. ACE2 is the receptor used by SARS-CoV-1 and SARS-CoV-2 to enter host cells and infect the body; therefore, upregulation of this receptor increases entry points for these viruses [71]. A study by Hui et al. [86] found that the MERS-CoV DPP4 receptor was upregulated in the lungs of smokers who suffer from chronic obstructive pulmonary disease. The upregulation of this receptor could potentially explain how patients with comorbid lung disease are at greater risk for MERS. In addition, smokers exhibit significantly reduced levels of serum vitamin D [88], which may prevent protective effects against respiratory illness and worsen clinical symptoms. Although smoking increases vulnerability to respiratory illness, studies have hinted at the potential benefits of nicotine when fighting COVID-19. Nicotine is a cholinergic agonist, important in the inhibition of proinflammatory cytokines such as TNF and IL-6. Thus, nicotine inhibition is being considered a potential therapy against the elevated IL-6 levels in COVID-19 patients [89].

There is also growing evidence suggesting a link between asthma patients, lung function, and vitamin D levels [72]. Asthma is caused by various underlying mechanisms and presents heterogeneously with assorted clinical manifestations [90]. Vitamin D deficiency paralleled the decrease in lung function seen in asthmatic children. While there have not been many studies investigating the risk of asthma for COVID-19 symptoms, the CDC has listed asthma as a risk factor for severe illness from COVID-19. A recent study analyzed data from the U.K. Biobank to collect comprehensive phenotypic data and longitudinally measured health outcomes through linkages to natural data sets found that participants with asthma had a significantly higher risk for severe COVID-19. It is interesting to note that nonallergic asthma specifically was significantly associated with severe COVID-19, while allergic asthma had no statistically significant association with the virus [91].

## 4. Therapies and Vaccines

A number of therapeutic options and vaccines are under investigation for effectiveness, with some being rolled out for public administration currently. As the knowledge base of viral mechanisms continues to grow, the plausible avenues of treatment will inevitably expand, ranging from targeting the symptomology at the level of RAS disturbances and inflammatory overactivation to passive immunotherapy and the repurposing of approved drugs. COVID-19 therapies currently in clinical trials are listed in Table 2.

Currently, the use of corticosteroids early in the treatment of COVID-19 symptoms has proven to be effective in hospitalized patients. A meta-analysis including 44 studies on corticosteroid use showed that there was a significant decrease in mortality upon corticosteroid use in treatment [92]. The study also determined that corticosteroid treatment decreased the requirement for and duration of mechanical ventilation. While corticosteroid treatment has shown to be effective, it also results in delayed viral clearance by 2–4 days and an increased risk of developing secondary infections.

The renin-angiotensin system (RAS) serves as a potential guide for understanding the pathogenesis of SARS-CoV-2 infections, particularly in patients with hypertension. Increased Ang II levels are observed in COVID-19 patients, hypertensive patients, and individuals with lung failure. Elevated angiotensin II levels can increase inflammatory cytokine levels, like IL-6, which may explain the link between inflammatory lung disease and COVID-19 [65]. One study involving COVID-19 patients with hypertension found that ACE-inhibitors and Ang II type 1 receptor blockers (ARBs) reduced inflammatory cytokine production and IL-6 levels in peripheral blood [93]. This study suggests that these agents may help reduce inflammation and slow COVID-19 pathogenesis in hypertensive patients.

Sarilumab, an IL-6 blocker, is currently under evaluation for its efficacy in treating pneumonia and hyper-inflammation associated with COVID-19. One study conducted in Italy did not report statistically significant differences between sarilumab and control groups in the treatment of severe COVID-19 [94]. Given that the median age of this study population was 56 years old and the higher mortality in older groups, a higher dose of sarilumab may be required for significant positive clinical outcomes in younger patients. Additionally, tocilizumab, a related drug, elicited adverse infectious reactions when administered at higher doses, presenting safety risks for patients. Such findings may be linked to IL-6 inhibitor immunosuppression, which can be negatively exacerbated at higher doses in humans. One study conducted at Union Hospital in Wuhan, China, assessed the effectiveness of interferon-alpha-2b (IFN-a2b) and arbidol on moderate cases of COVID-19 patients with viral pneumonia. IFN-a2b is an FDA-approved antiviral used to treat genital warts and hepatitis B, while arbidol is a broad-spectrum antiviral officially used in Russia and China for influenza treatment. Patients received IFN-a2b, arbidol, or a combination of both and were found to have significantly reduced circulating IL-6 and CRP levels [95]. IFN-a2b also significantly increased viral clearance of SARS-CoV-2 from the upper respiratory tract in patients. Although this was an uncontrolled study, it further supports the link between acute inflammatory biomarkers such as IL-6 and CRP with COVID-19 pathogenesis. IFN-a2b was also exhibited to have antiviral effects against SARS-CoV-1 during the 2003 outbreak in Toronto and warrants further evaluation in a randomized clinical trial [95].

Passive immunotherapy is a potential therapeutic option for treating MERS-CoV infections as an alternative to currently marketed drugs such as remdesivir. The National Institute of Allergy and Infectious Diseases sponsored a clinical trial that analyzed the effect of SAB-301 on MERS-CoV infection [96]. SAB-301 is a human polyclonal IgG antibody derived from transchromosomic cattle previously vaccinated by a MERS-CoV vaccine. SAB-301 was found to be a safe and tolerable therapy among non-infected subjects. Lopinavir (LPV) and ritonavir (RTV), marketed as Kaletra, are currently used for HIV patients as protease inhibitors and are considered broad-spectrum drugs for coronavirus treatment [97]. LPV-RTV administration has been shown to improve the clinical outcomes of patients experiencing MERS and SARS [98].

The MIRACLE study, an international project initiated in Saudi Arabia, has identified LPV-RTV and interferon-beta-1b (IFN-beta-1b) as effective treatments against MERS. LPV inhibits the MERS-CoV replication cycle, while RTV increases LPV serum concentration by inhibiting its metabolism [99]. This dual effect may be a primary mechanism that prevents MERS-CoV pathogenesis in humans. A retrospective study involving healthcare workers found that LPV-RTV administration using a post-exposure prophylaxis model significantly lowered MERS-CoV infection risk. Although additional clinical evidence is required to confirm the efficacy of LPV-RTV in MERS-CoV infection, this drug combination with IFN-beta is officially used in Korea for treatment [98]. Clinical effects of LPV or LPV-RTV on COVID-19 remain unknown.

Current psychiatric drugs may be a promising area of the off-label use or drug repurposing for COVID-19 treatment. The reCoVery study, a clinical trial based in France, will evaluate the potential immunomodulatory benefits of chlorpromazine (CPZ), a phenothiazine used for antipsychotic treatment [100]. CPZ in mice models is associated with increased IgM levels, as well as reduced IL-2, IL-4, IFN-alpha, TNF, and GM-CSF cytokines. Researchers on this project postulate that CPZ is clinically advantageous due to its higher bioavailability compared to antivirals. Additionally, CPZ is highly concentrated in both the lungs and the salivary glands, which may decrease overall SARS-CoV-2 viral load and human-human transmission. This study highlights the importance of discovering off-label use in drugs outside of traditional antivirals [100]. Given the highly collaborative efforts in academic laboratories and pharmaceutical companies worldwide, it is hoped that the host-pathogen interactions of existing CoVs in humans will be elucidated for the effective future development of vaccines and therapies.

### Vaccines

One of the critical ways of preventing future pandemics is by designing a vaccine model for scalable manufacturing, delivery, and global distribution. Many clinically administered antiviral vaccines are live attenuated vaccines, inactivated vaccines, or viral vector vaccines [101]. Naturally, most vaccine candidates for COVID-19 prophylaxis in clinical trials are one of these modern vaccine types (Table 3). The primary concern of development is ensuring vaccine accessibility in countries across the wealth spectrum, especially in lower-income countries. Nanotechnology platforms may serve as a promising area of research for designing safer, effective, and more globally accessible vaccines. Given their scale, viruses can be considered natural nanomaterials. Therefore, natural or synthetic nanoparticle technology may allow for the development of next-generation vaccines. DNA and RNA-based vaccines, for example, have emerged in recent clinical trials for COVID-19 vaccination due to their ability to evoke a robust immune response, including humoral and CD4^+^ and CD8^+^ T-cell responses. While DNA vaccines are more stable than mRNA vaccines, DNA vaccines can cause insertional mutagenesis. Moderna, a U.S.-based biotechnology company, was the first company to develop a lipid nanoparticle mRNA-based vaccine, the first vaccine to reach clinical trial development in the U.S. Inovio Pharmaceuticals is currently developing a DNA-based vaccine that entered phase 1 clinical trials in April 2020. Peptide-based vaccines and subunit vaccines, both more modern vaccine formulations, are also being engineered in academic laboratories and scientific companies worldwide as potential vaccine platforms [101]. Such rapid developments in next-generation vaccine production through nanotechnology platforms provide hope of eventually creating an effective COVID-19 vaccine. Overall, pinpointing the relative impact of humoral and cellular immune responses will inform both vaccine and therapy development moving forward. Due to emergency conditions, several vaccines have been authorized for administration, as listed in Table 3. Vaccines approved for emergency use in the U.S. include the Moderna and Pfizer vaccines. According to the CDC, 50.1% of the total U.S. population is fully vaccinated, and 58.5% have at least one dose [102].

Very few studies on potential SARS-CoV-1 vaccines and MERS-CoV vaccines have advanced past the preclinical stage [103]. Various projects have been conducted to assess the safety and effectiveness of DNA-based and viral vector vaccines against MERS-CoV (Table 4). Following intramuscular electroporation of GLS 5300, anti-MERS seroconversion was observed in over 90% of patients [104]. Additionally, administration of GLS 5300 elicited T-cell responses against MERS-related spike glycoproteins. These results suggest that further clinical trials analyzing the effect of GLS 5300 in patients are warranted. Several pre-existing vaccine models for current infectious diseases are under investigation for possible use against MERS. One such vaccine, modified vaccinia virus Ankara (MVA), is a viral-vector-based vaccine that has undergone phase 1 clinical trial analysis to prevent MERS-CoV viral infection. MVA-MERS vaccination was associated with B- and T-cell responses in patients [105]. Notably, T-cell response was most prominent during MERS spikes post-vaccination, suggesting that T cells play a more significant role in viral clearance than B cells. Anti-MERS-CoV antibody administration was shown to mitigate MERS-CoV viral infection in mice expressing the DPP4 receptor and may be worth further consideration in human studies. ChAdOx1, an additional viral vector vaccine, exhibited protection against MERS-CoV infection via neutralizing antibodies and CD8^+^ T cells in transgenic mice [104].

A table of vaccines is currently in development for SARS-CoV-1 and MERS-CoV. An inactivated vaccine is currently in a phase 1 clinical trial and has been shown to contribute to 100% seroconversion following two vaccine doses. Another vaccine being developed by the NIH is a recombinant DNA vaccine that is in phase 1 and has shown to produce a CD4^+^ response. GLS 5300 and ChAdOx1 are currently being developed for MERS-CoV. Both are currently in phase 1 clinical trials and have been shown to provide immune protection for at least a year after vaccination.

## 5. Conclusions

Pre-existing conditions, notably pulmonary and cardiovascular diseases, have been explored in various studies to uncover the pathophysiological mechanism of COVID-19. Additionally, factors such as race, biological sex, BMI, and nutritional deficiencies serve as potential predictors of disease severity and clinical outcomes. Given that older populations are considered high risk, individuals within this demographic are continuously recruited for research and treatment purposes. Despite research efforts on novel COVID-19 therapeutics, no specific FDA-approved treatment has been identified to manage its symptoms in humans effectively. The evidence provided in this review highlights patients with high-risk factors, especially those with elevated IL-6 levels, may particularly benefit from clinically approved neutralizing antibodies against IL-6R. This can potentially attenuate the hyper-inflammatory response leading to respiratory failure and viral spread within the host.

Additionally, current clinical trials have shifted their primary focus to finding vaccines and therapeutics marketed and/or broad-spectrum, meaning effective against various types of viruses. Fortunately, the genome sequence homology between SARS-CoV-1 and 2 allows for the repurposing of vaccines and treatments, which provides a more efficient means of mitigating viral spread. It is essential to mention the delay within the global scientific community in translating in vitro findings to clinical applications despite the emergence of three coronavirus outbreaks since the early 2000s. When SARS and MERS emerged, potential vaccines in development lacked financial investment to reach the market due to the low infection numbers [101]. Given the rapid genetic recombination associated with these viruses, it was no surprise that a novel coronavirus-related pathogen emerged unexpectedly. Therefore, therapeutic development against COVID-19, for example, may not be helpful against a future coronavirus that may appear. This likely explains why many researchers are repurposing currently marketed drugs, such as Lopinavir and Ritonavir, to mitigate viral spread. SARS-CoV-2 is particularly unique due to its transmission from asymptomatic carriers. Therefore, the WHO’s guidelines on preventing SARS-CoV-2 spread must continue to be implemented by the general global population. Additionally, the potential protective effects of vitamin D against COVID-19 pathogenesis are worth further analysis. As discussed above, ensuring sufficient vitamin D levels in COVID-19 patients might minimize symptom severity.

## 6. Gaining Perspective: Lessons from the Existing Coronaviruses

Overall, the risk factors mentioned in this paper highlight the interconnected nature of COVID-19 pathogenesis. For example, obesity can decrease vitamin D bioavailability and further inhibit macrophage modulation of the immune response. An important focus of vaccine and therapeutic development should be designing animal models that genetically represent the diversity of the human population. Additionally, the natural hosts of current CoVs are animals and viral surveillance of wild animals may help prevent a future outbreak. Beyond genetic considerations in preventing coronavirus spread, additional lifestyle and pre-existing factors should be further analyzed in future studies. Our commentary on therapies and vaccines for COVID-19 highlights the rapid response by the scientific community to the ongoing pandemic and provides hope that viral spread will be contained in the foreseeable future. It is important to note that prevention of future viral pandemics will require more than vaccine and therapeutic development. From a racial standpoint, the Latinx population is also at an increased risk of COVID-19 infection due to immigration status on healthcare access [106]. For example, undocumented Latinx individuals are often reluctant and fearful to seek healthcare services due to their immigration status. Thus, the structural barriers to healthcare have been exposed during this time given the emerging racial discrepancies in COVID-19 infection and mortality rate, particularly in the U.S. communities of color who struggle with pre-existing, ongoing racial, and economic discrimination in America are disproportionately affected by COVID-19. Thus, improving the environments in which Black and Brown communities reside is imperative in preventing viral outbreaks in such communities and mitigating the impact of stress on their immune response. Finally, this has set a precedent for how global health challenges must be urgently addressed by institutions of power worldwide, regardless of political, financial, and geographical differences.

A recent rise in COVID-19 cases from the November of 2020 through May of 2021 has raised concern for emerging variants of SARS-CoV-2. Specific countries impacted by this “second wave” include the U.K., Brazil, South Africa, and India. These variants have been shown to emerge independently of one another in different regions [107]. Variants of concern in these countries include B.1.1.7 (501Y.V1) in the U.K., B.1.1.28.1 (501Y.V3) in Brazil, and B.1.351 (501Y.V2) in South Africa [108]. These variants have spread to countries such as India, where the 501Y.V1 strain seen in the U.K. is the most prevalent of the three. Other strains of concern in India currently include B.1.617 and B.1.618, with B.1.617 being the more predominant of the two [109]. B.1.617 contains three main mutations located in genes encoding the viral spike protein: L452R, E484Q, and P681. These mutations are all thought to play roles in conferring viral resistance to neutralizing monoclonal antibodies, increasing viral load by altering host protein function, and increasing infectivity by enhancing spike protein-ACE2 binding and S1-S2 spike protein cleavage [108,109]. Another variant, N501Y, has been shown to enhance binding affinity between viral S protein and ACE2 receptor by the formation of a π-π interaction with Y44 of ACE2 [107]. Other variants with mutations within the receptor-binding domain of the spike protein may be able to bind to alternate host receptors and may also avoid binding by vaccine-induced host immunoglobulins. Clinical concerns of these emerging variants are that infectivity, transmissibility, and viral load of the virus are all increasing; RT-PCR tests are designed to detect the original spike protein and may fail to detect new strains in patients, possibly contributing to an increase in spread by non-symptomatic individuals; and there is potential for reduced effectiveness of vaccines due to decreased binding by antibodies.

Another point of concern in the recent rise of cases in countries such as India is the rising number of cases of mucormycosis, otherwise known as “black fungus.” As of May 2021, over 100 cases of COVID-19-associated mucormycosis (CAM) have been reported, with at least 82 of the cases in India [97,108]. Other sources report that India contributed to approximately 71% of the CAM cases between December 2019 and April 2021 [110]. Both rhinocerebral mucormycosis and nasal sinus mucormycosis have been observed in COVID-19 patients. Mucormycosis is an opportunistic fungal infection caused by several global fungal species, including Rhizopus species and Mucor species, that primarily infects immunocompromised patients [111]. Infection by mucormycosis is not considered to be contagious or transmissible from person to person. Infection by the fungus typically occurs from the inhalation of mucormycete spores from the environment. Although many cases have been detected specifically in India, mucormycosis-causing fungi are found globally. Immunocompromised individuals have an increased risk of developing mucormycosis. Risk factors for mucormycosis include diabetes, cancer neutropenia, organ transplant, long-term corticosteroid use, and skin injury. CAM cases have been observed predominantly in males, with males reaching about 80% of CAM cases [109]. Research on 101 observed CAM cases demonstrated that 80% of the patients had pre-existing diabetes mellitus, and corticosteroid treatment was given to treat COVID-19 in 76.3% of these patients. Other risks for developing mucormycosis involve the healthcare setting: poor air filtration, non-sterile medical devices, construction, and water leaks. While the incidence of black fungus cases is increasing as new variants emerge, the cases of mucormycosis seem to be due to increasing cases in people with the key risk factors rather than as a product of the new variants themselves. The critical risk factors pre-existing diabetes mellitus, COVID-19, and corticosteroid treatment, when combined, significantly increase one’s risk of developing a mucormycosis infection.

## Figures and Tables

**Figure 1 vaccines-09-00938-f001:**
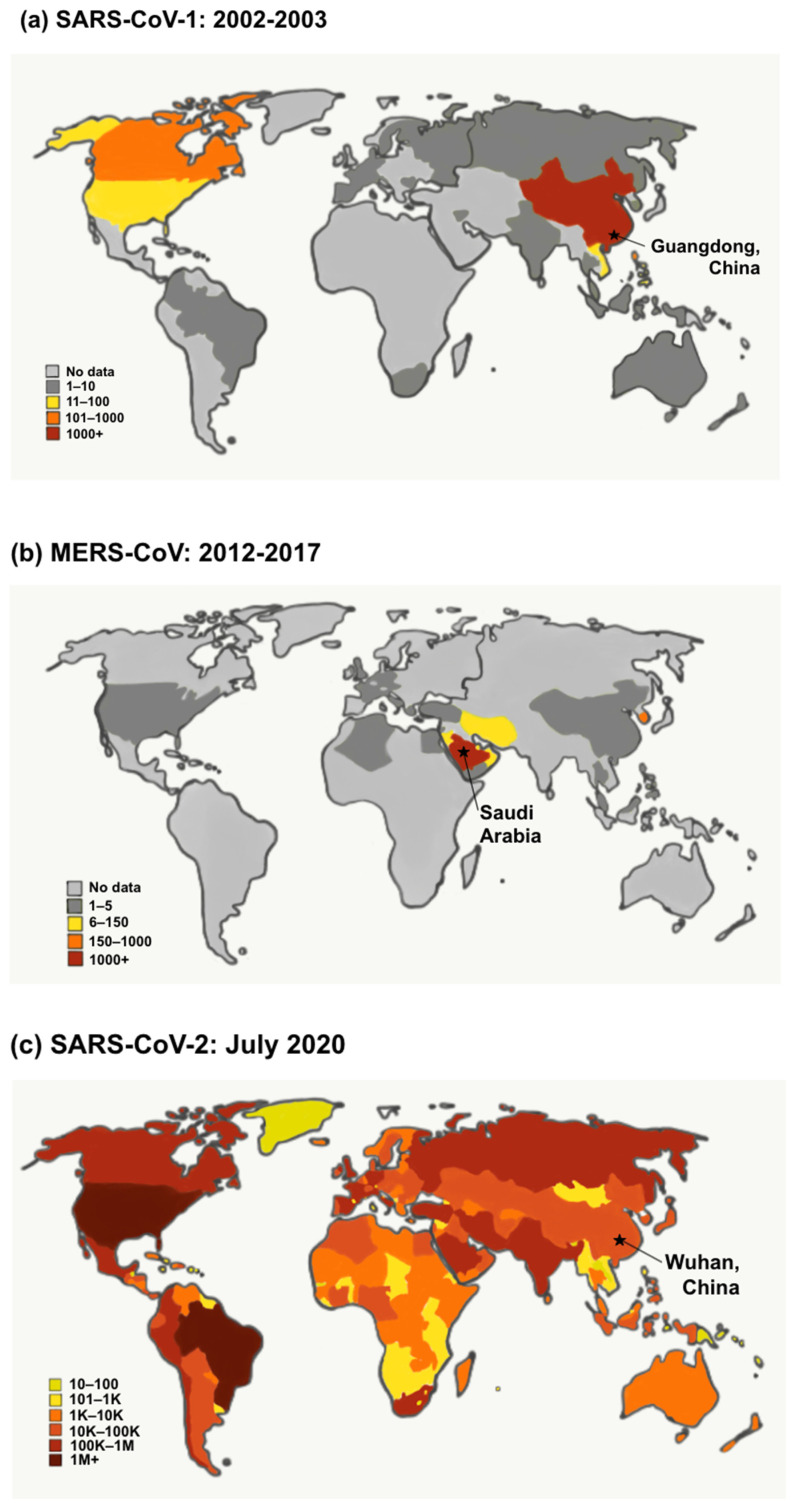
The confirmed number of global cases for (**a**) SARS-CoV-1, (**b**) MERS-CoV, and (**c**) SARS-CoV-2. (**a**,**b**) Figures based on data from the World Health Organization. (**c**) Figure based on data from the John Hopkins COVID-19 report data. Note that the color scales vary for each world map. (**a**) The SARS-CoV-1 outbreak was reported in Guangdong, China, in 2002 from an unknown animal species, most likely a bat. The highest number of cases occurred in China (1000+), as shown in red on the map. Regions marked in orange had 101–1000 reported cases, yellow had 11–100 reported cases, and green had 1–10 reported cases. The regions in light gray have no data on reported SARS-CoV-1 cases. (**b**) The MERS-CoV outbreak was identified in Saudi Arabia in an animal believed to be bats but is also endemic in the dromedary camel, which is the animal reservoir responsible for human infection. The largest number of MERS-CoV cases occurred in Saudi Arabia (1000+), as shown in red on the map, and the Republic of Korea (150–1000), as shown in orange on the map. Areas with 6–150 cases are shown in yellow, 1–5 cases are shown in green, and no data on reported cases are shown in light gray. (**c**) The SARS-CoV-2 pandemic was discovered in Wuhan, China, in 2019 from an unknown animal origin, most likely a bat. The SARS-CoV-2 map shows the number of cumulative confirmed global cases as of December 2020. The color scale for the SARS-CoV-2 map shows a more drastic change in the number of cases as colors change. The darkest red color shows areas at more than 1 million cumulative confirmed cases. The color scale for the map reads as follows: dark red = 1 million + cases, red = 100,000 to 1 million cases, red-orange = 10,000 to 100,000 cases, orange = 1000 to 10,000 cases, yellow = 101 to 1000 cases, green = 1 to 100 cases, light gray = no data reported.

**Figure 2 vaccines-09-00938-f002:**
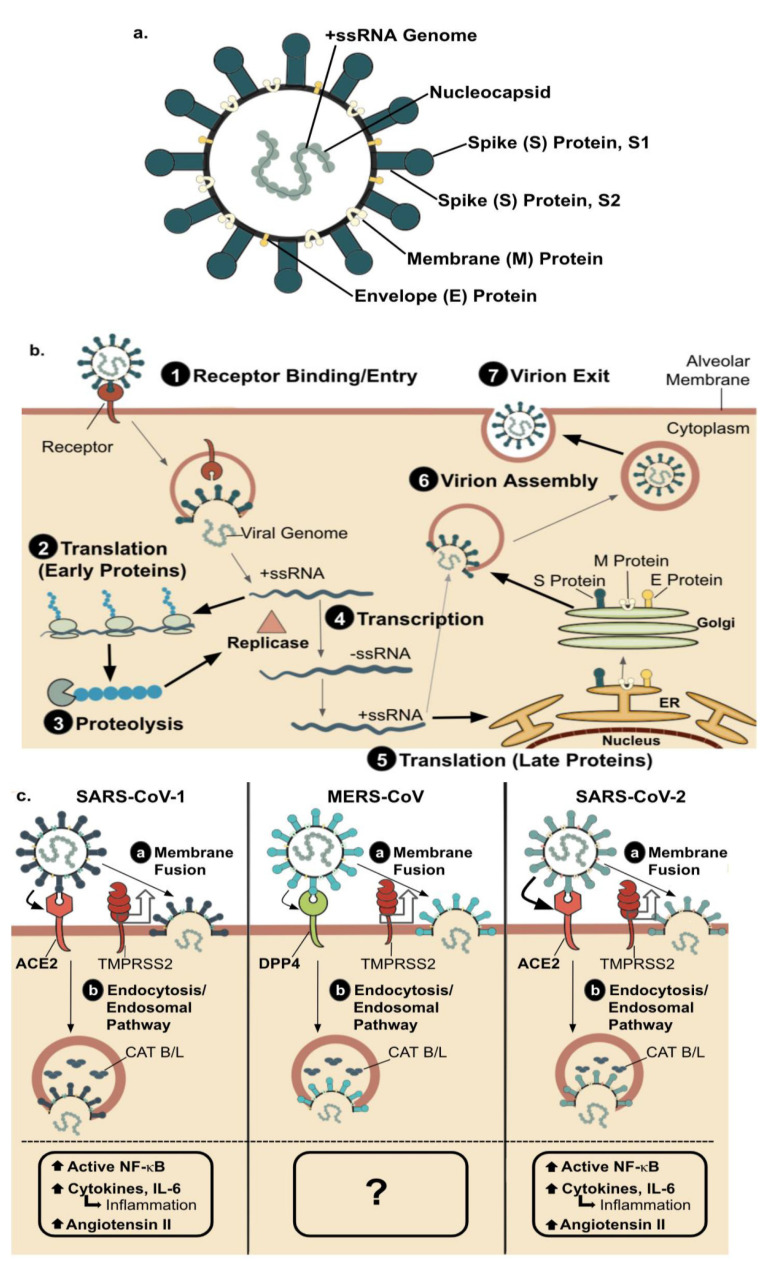
(**a**)**.** Coronavirus structure. The coronavirus structure includes the (+) ssRNA genome contained within the nucleocapsid surrounded by an envelope. The envelope contains envelope (E) protein, membrane (M) protein, and spike (S) protein that has two subunits, S1 and S2 (Li, 2016). The envelope protein (E), the third of four structural proteins, plays a role in the assembly and release of the virus. The membrane protein (M) promotes membrane curvature and viral assembly. S1 contains the RBD and is responsible for binding to the host’s cellular receptor, while S2 aids in the fusion and entrance process of the virus into the cell. The nucleocapsid protein (N) is contained within the envelope, and it supports viral replication by packaging the genome into the virion particle and antagonizing IFN-mediated host immunity. (**b**). General life cycle of a coronavirus. (1) the virus binds via the S1 subunit receptor-binding domain to the appropriate receptor (shown in (**c**) for each respective coronavirus) and enters the cell either through direct fusion with the cell membrane or through the endocytosis/endosomal pathway, as shown in the figure. Note that the figure depicts the alveolar membrane, as coronavirus receptors are predominantly found on the alveolar membrane in the lower respiratory tract. The end result of receptor binding and entry is release of the +ssRNA viral genome in the host cell cytoplasm. (2) The +ssRNA can be directly translated by host cell ribosomes. Translation of early proteins occurs first, producing a polypeptide chain that needs to be cleaved to be activated. (3) The polypeptide chains produced in translation are cleaved and activated in a step called proteolysis. One of the activated proteins produced includes replicase. (4) An RNA replicase-transcriptase complex forms to create -ssRNA strands from the +ssRNA. The -ssRNA strands are used as templates to create more genomic +ssRNA, which can be packaged into virions and also (5) translated to produce late viral proteins. The late viral proteins include structural proteins S, M, and E, which are incorporated into the virion in virion assembly (6). (7) The virions then exit the host cell through budding. (**c**). Coronavirus receptor binding and entry: SARS-CoV-1, MERS-CoV, and SARS-CoV-2. SARS-CoV-1, MERS-CoV, and SARS-CoV-2 differ in receptor type, receptor-ligand affinity, and downstream effects. This figure depicts the alveolar epithelial membrane. Both SARS-CoV-1 and -2 bind to ACE-2 receptor in the alveolar epithelial membrane. The RBD residues in the S glycoprotein of SARS-CoV-2 binds to the ACE2 receptor with 10 to 20 times higher affinity when compared to that of SARS-CoV-1. The difference in RBD affinity for the ACE2 receptor is demonstrated in the figure with a bigger arrow showing SARS-CoV-2 binding to the ACE2 receptor versus SARS-CoV-1 binding the ACE2 receptor. MERS-CoV binds to the DPP4 receptor. DPP4 receptors show a preferential spatial location in Alveolar epithelial cells (depicted in the figure above), vascular endothelium (lymphatics), and pleural mesothelia; thus, MERS is characterized as a lower respiratory tract disease. All three of the depicted coronaviruses can release the RNA viral genome into the cell via (**a**) membrane fusion mediated by protein TMPRSS2 or (**b**) endocytosis and the endosomal pathway mediated by CAT B/L. Downstream effects of SARS-CoV-1 and -2 are similar, including increased activation of the NF-κB pathway, increased cytokines including IL-6 promoting inflammation, and accumulation of angiotensin II due to reduced ACE2 receptor availability [12]. The downstream cellular effects of MERS-CoV are not well characterized.

**Figure 3 vaccines-09-00938-f003:**
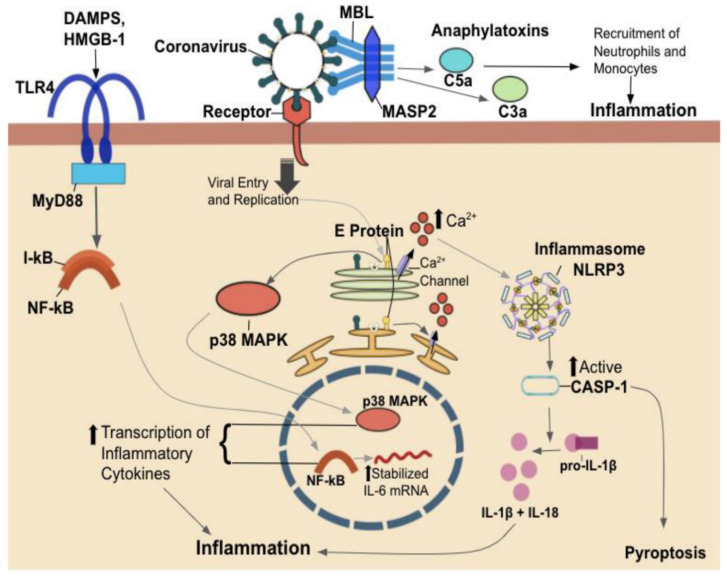
Possible outcomes of coronavirus replication in alveolar epithelial cells: inflammasome, TLR, lectin. Inflammasome: coronavirus receptor binding, entry, and replication results in an increased presence of viral envelope (E) protein in the cell. E protein contains a PDZ-binding motif (PBM), which interacts with host cellular proteins. E protein forms Ca^2+^ channels in the endoplasmic reticulum Golgi apparatus intermediate compartment. Increased cytoplasmic Ca^2+^ levels alter the host cell’s homeostatic state, inducing activation of cytosolic innate immune signaling receptor NLRP3 (NOD-, LRR-, and pyrin domain-containing 3) inflammasome. Upon NLRP3 activation, caspase-1 (casp-1) is activated, which then goes on to cleave and activate proinflammatory cytokines such as pro-IL-1β into active IL-1β and IL-18. NLRP3 activation also induces pyroptosis, a caspase-1-dependent proinflammatory cell death. E protein’s PBM also interacts with syntenin proteins to trigger activation of the p38 MAPK pathway. TLR: TLR4, present on alveolar epithelial cells, is activated by damage-associated molecular patterns (DAMPS), including HMGB-1, which is found in patients infected by coronavirus. MyD88 acts as an adaptor for TLR receptors. TLR4 activation induces downstream NF-B activation, which acts as a transcription factor for proinflammatory cytokines. TLR4-mediated NF-B activation also induces the stabilization of IL-6 mRNA, allowing for more IL-6 production. Lectin: SARS-CoV spike (S) protein interaction with serum mannose-binding lectin (MBL) induces activation of MBL-associated serine protease 2 (MASP-2). The MBL-MASP2 complex activates a signal cascade that produces anaphylatoxins C3a and C5a, which recruit neutrophils and monocytes, promoting inflammation.

**Figure 4 vaccines-09-00938-f004:**
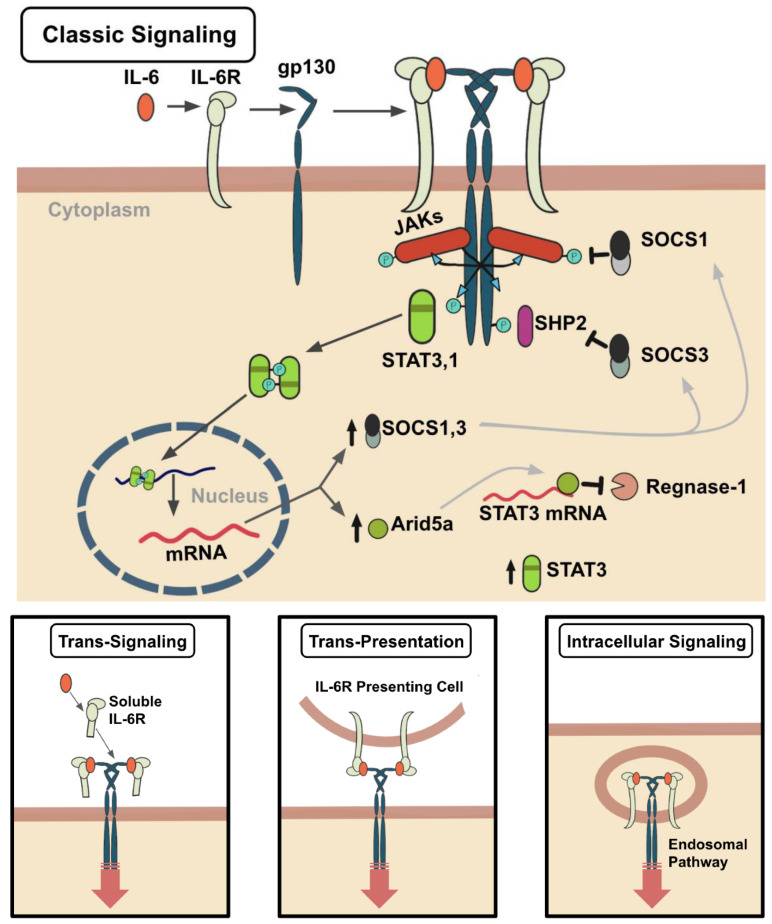
IL-6 signaling pathways: classical signaling via the JAK/STAT pathway, trans-signaling, trans-presentation, and intracellular signaling. Once IL-6 is produced, it binds to its specific membrane receptor, IL-6R (or CD126), inducing homodimerization of the signaling receptor component gp130 (or CD130) on the surface of cells such as hepatocytes, neutrophils, monocytes, activated B cells, and CD4 T cells (classic signaling). This homodimerization then activates cytoplasmic Janus kinases (JAKs: JAK1, JAK2, and TYK2), which phosphorylate tyrosine residues in the cytoplasmic region of gp130. The downstream signaling molecules, signal transducer, and activator of transcription 3 (STAT3), STAT1, and SH2 domain-containing protein-tyrosine phosphatase 2 (SHP2) are recruited to the tyrosine-phosphorylated motifs of gp130. These signaling molecules themselves are then phosphorylated by the aforementioned JAKs and thus translocated to the nucleus, producing the transcriptional output. Similar to how IL-6 production is controlled by both stimulatory and inhibitory components, this is also the case with respect to regulating the IL-6 intracellular transduction signal. Specifically, STAT3 has been shown to have both a positive and negative effect on the transduction pathway. In one role, it stimulates negative feedback molecules, suppressors of cytokine signaling 1 (SOCS1) and SOCS3 proteins. SOCS1 binds to and inhibits JAK, while SOCS3 inhibits the Ras-MAP kinase pathway induced by SHP2. In another role, STAT3 induces Arid5a, which selectively binds to STAT3 3′UTR and stabilizes the positive STAT3 signal. Thus, these molecules also provide a critical means of regulating the response to IL-6 and the prevention from pathologies associated with IL-6 dysregulation. IL-6 does not strictly follow the classic signaling pathway, as not all IL-6 receptors are membrane receptors. In a process called trans-signaling, IL-6 can bind to the soluble form of IL-6R (sIL-6R) in the serum. The IL-6/sIL-6R complex then activates gp130-expressing cells, even if cells lack membrane IL-6R. This may, in part, explain the pleiotropic effects of IL-6, given that gp130 is expressed in many more cells than those who express membrane IL-6R. Furthermore, cells expressing only IL-6R can present IL-6 to cells expressing only gp130 and activate the signaling pathway, in what is called trans-presentation. Lastly, extracellular IL-6 can be taken up by IL-6R and translocated to the endosomal compartment by endocytosis, where it activates gp130.

**Table 1 vaccines-09-00938-t001:** Genetic susceptibility based on HLA allele composition toward respiratory infections.

Disease	Increased Risk Allele	Decreased Risk Allele
MERS	HLA-DRB1*11:01 HLA-DQB1*02:02	
SARS-CoV-1	HLA-B*46:01 HLA-B*5401 HLA-B*0703 HLA-DRB1*03:01 HLA-DRB1*12:02	HLA-DR0301
SARS-CoV-2	HLA-B*46:01 HLA-DRB1*01:01 HLA-BRB1*03:02 HLA-BRB1*03:03 HLA-A*25:01 HLA-C*01:02 HLA-C*07:29 HLA-B*15:27	DRB1*10:10 DRB1*01:01 DRB1*01:04 HLA-B*15:03 HLA-A*02:02 HLA-B*15:03 HLA-C:12:03

HLA alleles that have been shown to be associated with a change in risk toward various respiratory infections. HLA-DRB1*11:01 and HLA-DQB1*02:02 were associated with disease but not outcome in Saudi patients with MERS. HLA-B*46:01 and HLA-B*5401 were associated with decreased peptide presentation, leading to higher risk in Taiwanese patients with SARS-CoV-1. HLA-B*0703, HLA-DRB1*03:01, and HLA-DRB1*12:02 were found at higher prevalence rates in Chinese patients infected with SARS-CoV-1. HLA-DR0301 showed resistance toward SARS-CoV-1 in a population of Taiwanese patients. HLA-B*46:01, HLA-DRB1*01:01, HLA-BRB1*03:02, and HLA-BRB1*03:03 all showed weak peptide binding, increasing the risk of SARS-CoV-2 infection in a global population. HLA-A*25:01 and HLA-C*01:02 showed weak peptide binding, increasing the risk of infection in an in silico analysis. HLA-C*07:29 and HLA-B*15:27 were found at an increased prevalence in a population of chinses patients infected with SARS-CoV-2. DRB1*10:10, DRB1*01:01, DRB1*01:04, and HLA-B*15:03 were associated with increased peptide presentation, decreasing the risk of infection to SARS-CoV-2 in a global population. HLA-A*02:02, HLA-B*15:03, and HLA-C:12:03 were found also associated with increased peptide presentation in an in silico analysis.

**Table 2 vaccines-09-00938-t002:** COVID-19, SARS, and MERS therapies in development.

Name	MOA	Clinical Outcome(s)	Comments	Trial
COVID-19				
Tocilizumab	Humanized monoclonal IL-6R antibody	Decreased patient temperature, oxygen intake, and CRP levels	Currently used to treat rheumatoid arthritis	Phase 2/3
Hydroxychloroquine	Unknown	Not significantly more effective than standard of care	Used to treat malaria and rheumatoid arthritis	Phase 3
Favipiravir	Viral RdRp inhibitor	Increased viral clearance and improved chest CT scans	Effective against Ebola virus and influenza	Phase 2/3
LPV/RTV	LPV: Protease inhibitor RTV: LPV metabolism inhibitor	Eliminated detectable SARS-CoV-2 viral titers (44% of patients)	No detectable side effects found	Phase 2
Arbidol	Viral entry inhibitor	Eliminated detectable SARS-CoV-2 viral titers Delayed the progression of pulmonary lesions	Currently used to treat prophylaxis and influenza in Russia and China	Phase 4
SARS				
Remdesivir	Viral RdRp inhibitor	Reduces pulmonary SARS-CoV viral load	Broad-spectrum antiviral	Preclinical
LPV/RTV	LPV: Protease inhibitor RTV: LPV metabolism inhibitor	Reduces SARS-CoV viral load, mortality, and intubation rate	Possibly most effective as an initial therapy against infection	Preclinical
MERS				
REGN 3048 and REGN 3051	Humanized mAbs	Clinical trial results not yet published	None	Phase 1
LPV/RTV and IFN-beta-1b	LPV/RTV: Protease inhibitors IFN-beta-1b: Antiviral protein activator	Contributed to a 40% decrease in early stage infection risk	None	Phase 1
SAB-301	Human polyclonal IgG	Clinical trial results not yet published	No reported tolerance issues	Phase 1

A summary of the current proposed therapies for the treatment of COVID-19, SARS, and MERS. Tocilizumab is a monoclonal IL-6R antibody that is undergoing phase 2/3 trials and has been effective in decreasing patient temperature, oxygen intake, and CRP levels. Hydroxychloroquine, an anti-malarial, is in phase 3; however, it has not been shown to be more effective than the standard of care. Favipiravir is a viral RdRp inhibitor in phase 2 and has been shown to eliminate detectable SARS-CoV-2 viral titers. Arbidol is in phase 4, and its mechanism of action inhibits viral entry and has shown to eliminate viral titers and delay the progression of pulmonary lesions. In regard to SARS, both remdesivir and LPV/RTV are in preclinical trials and have been shown to reduce viral load. REGN 3048, REGN 3051, and SAB-301 are all in phase 1 clinical trials for the treatment of MERS without any published results. LPV/RTV is a protease inhibitor, and IFN-beta-1b is an antiviral protein activator. When used in combination, they have been shown to reduce infection risk of MERS by 40% in a phase 1 clinical trial. Abbreviations: MOA = mechanism of action; IL-6R = interleukin 6 receptor; RdRp = RNA-dependent RNA polymerase; CRP= C-reactive protein; LPV/RTV = lopinavir-ritonavir; GI = gastrointestinal; mAbs = mouse antibodies; IFN = interferon; RBD = receptor-binding domain; IgG = immunoglobulin G Note: “Trial Phase” indicates the overall phase in development of the designated drug for the specified coronavirus; clinical outcomes reflect preclinical findings. The studies listed have no current updates on clinical trial results, which speaks to the delay in translating in vivo findings into tangible clinical applications.

**Table 3 vaccines-09-00938-t003:** SARS-CoV-2 vaccines in development.

Name	Developer(s)	Vaccine Type	Trial Phase
* mRNA 1273	Moderna and NIAID	mRNA vaccine	Phase 3
* ChAdOx1	University of Oxford and AstraZeneca	Non-replicating viral vector	Phase 2b/3
CoronaVacc	Sinovac	Inactivated vaccine	Phase 3
N/A	Beijing Institute of Biological Products and Sinopharm	Inactivated vaccine	Phase 1/2
N/A	Wuhan Institute of Biological Products and Sinopharm	Inactivated vaccine	Phase 1/2
N/A	Institute of Medical Biology, Chinese Academy of Medical Sciences	Inactivated vaccine	Phase 1
N/A	Novavax	Subunit vaccine	Phase 1/2
Ad5-nCoV	CanSino Biological Incorporation, Beijing Institute of Biotechnology and Canadian Center for Vaccinology	Non-replicating viral vector (adenovirus type 5)	Phase 1 and phase 2
N/A	Shenzhen Geno-Immune Medical Institute	Non-replicating viral vector	Procedure 1: Phase 1/2Procedure 2: Phase 1
INO-4800	Inovio Pharmaceuticals	DNA vaccine	Phase 1
N/A	Symvivo	DNA vaccine	Phase 1
* BNT162	BioNTech, Pfizer, and Fosun Pharma	RNA vaccine	Phase 1/2
* COVAXIN	Bharat Biotech	Inactivated vaccine	Phase 3

Currently, mRNA 1273 and COVAXIN are in a phase 3 clinical trial as an mRNA vaccine and inactivated vaccine, respectively. ChAdOx1 is a non-replicating viral vector that is in phase 2b/3. CoronaVacc is an inactivated vaccine in a phase 3 clinical trial. In addition, there are three other inactivated vaccines in phase 1 or phase 1/2 clinical trials. There are two non-replicating viral vector vaccines, one of which is Ad4-nCoV, which are in phase 1 and phase 2 clinical trials. Two DNA vaccines, including INO-4800, are in phase 1 clinical trials. BNT162 is an RNA vaccine in a phase 1/2 clinical trial. * Approved for emergency use.

**Table 4 vaccines-09-00938-t004:** SARS-CoV and MERS-CoV vaccines in development.

Name	Developer	Vaccine Type	Clinical Outcome(s)	Comments	Trial Phase
SARS-CoV					
N/A	Sinovac	Inactivated vaccine	Contributed to 100% seroconversion following 2 vaccine doses of 16 SU	Short-term systemic adverse events (all subjects)	Phase 1
N/A	NIH, NIAID (Vaccine Research Center)	Recombinant DNA vaccine	CD4^+^ T-cell response (All subjects) CD8^+^ T-cell response (20% of subjects)	Mild systemic reactions observed (~50% of subjects)	Phase 1
MERS-CoV					
GLS 5300	Inovio Pharmaceuticals, Inc.	DNA vaccine	Associated with seroconversion and nAb development against MERS-COV viral infection	Immune protection lasted for one year after vaccination	Phase 1
ChAdOx1 MERS	Oxford University	ChAdOx1 vector	Associated with immune protection, seroconversion, and T-cell response	Immune protection lasted for a year after vaccination	Phase 1

Abbreviations: nAb = neutralizing antibody; NIH = National Institute of Health; NIAID = National Institute of Allergy and Infectious Disease; MVA = modified vaccinia virus Ankara; ChAdOx1 = chimpanzee adenovirus.

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
