# Peer review of "Risk Factors Associated with the Clinical Outcomes of COVID-19 and Its Variants in the Context of Cytokine Storm and Therapeutics/Vaccine Development Challenges"

_vaccines, 2021, doi:10.3390/vaccines9080938_

Round 1
Reviewer 1 Report
This is a very well written and educational review of the state of the art in understanding and managing COVID-19. This very comprehensive review makes extensive use of references and will be a good resource to researchers seeking to quickly understand COVID-19. This reviewer has the following minor concerns:
1. Section 4.1 on vaccines discusses how all COVID-19 vaccines are still in trials, and surprisingly ignores the fact that vaccines have been rolled out to much of the world under emergency use authorizations. Over 56% of the United States is now vaccinated. Although table 3 does list the current trial phase of vaccine candidates, it does not include which vaccines are approved for emergency use in the US and elsewhere. Also it does not talk about production, distribution, or storage complexities (e.g., temperatures for mRNA vaccines) in the vaccines. Some material on this should be included.
2. Reference 102, also in section 4.1, is used to support the statement that few vaccines have advanced beyond the pre-clinical stage. However, this was published in March 2020 and is is therefore very outdated.
3. Vitamin A deficiency is introduced for the first time in section 5, which is labeled “conclusions”. New concepts should not be introduced in a conclusion section.
4. In section 3.1, the manuscript overstates the significance of the risk of severe COVID being increased by psychological stress, at least according to the reference (#41). The reference indicates that psychological stress might play a role among many other risk factors, therefore this reference does not emphasize psychological stress as the primary indicator of severe disease. Other portions of the reference discuss systemic and societal problems leading to poorer outcomes, and these should be emphasized more than the potential minor role of psychological stress, unless other references can be cited.
5. In section 3.2.3, the discussion on whether A and AB blood types actually have significantly different COVID outcomes compared to type O is difficult to understand. Please revise to explain more clearly.
6. Section 3.3.1, lines 558-571 - It is difficult to understand what the authors are trying to convey about vitamin D’s downregulation of some immune functions.
7. Line 639, reference 92 appears to be the wrong reference as it has no mention of asthma in the referenced manuscript.
8. Section 4 does not discuss corticosteroids as a therapy for COVID-19, but this has proven one of the most effective therapies if COVID is treated early in hospitalization. Please add corticosteroids to the section.
Author Response
- We added discussion on percentage of vaccinated population in the US and vaccines that are currently authorized for use in the US. Discussion on production, distribution, or storage complexities is not included because we feel that is beyond the scope of the section’s focus.
- We changed to focus primarily on SARS-CoV-1 and MERS-CoV vaccines of which very few vaccines have advanced past the clinical stage.
- The section on Vit A deficiency was removed as it was new information in the conclusions section and seemed randomly placed.
- This part of the paper discusses psychological stress among the African American population as a contributing factor to an increased risk for COVID-19 infection and does not focus on it as a primary indicator of severe disease. The psycholocial stress is merely a point of discussion and consideration within the overall “gender and race” section of the paper.
- We revised the section and attempted to clarify the discussion on A and AB vs O blood types.
- We tried to make the words flow better to make it easier to understand.
- Fixed the reference.
- We added corticosteroid discussion to section 4.
Reviewer 2 Report
In the manuscript titled “Risk factors associated with the clinical outcomes of COVID-19 and its variants in the context of cytokine storm and therapeutics/vaccine development challenges” by Jain, et.al., the author gave a very comprehensive review of how the various risk factors affecting the current COVID-19 pandemic and how the knowledges regarding the risk factors giving the directions of generating therapeutics and vaccines. The authors also gave a description of the virology of SARS-CoV-2 and the immunological response to elucidate how the virus really works in the cell and the clinical manifestation and progression of COVID-19. The authors did a clinical data-based analysis regarding the various risk factors and how those risk factors are really making people vulnerable to the disease. This review also analyzed the latest approved vaccines. And this manuscript described a perspective on different variants and associated possible fungal infection (like black molds) towards the added mortality in several places.
Overall, this manuscript gave a compiled information on COVID-19 and its variants. This reviewer believes this manuscript is good for acceptance in Vaccines.
Author Response
This reviewer did not suggest edits to the paper.
Reviewer 3 Report
Very reliable and written review of the structure and function of the SARS CoV 2 virus and its comparison with the MERS virus and SARS CoV 1. A well-written chapter on the influence of environmental factors on infection. The authors focused on vitamin D deficiency, nicotine, and obesity. One can argue with this choice, as there are already more environmental factors described in the literature, e.g. changes in the intestinal microbiota or diseases (e.g. diabetes) which can have a critical influence on the course of the disease.
Author Response

(The authors gave the same response as above.)

Reviewer 4 Report
Hanna et al., has submitted the review entitled “Risk factors associated with the clinical outcomes of COVID-19 and its variants in the context of cytokine storm and therapeutics/vaccine development challenges”.
Although a surplus of literature reviews are available on “Risk factors associated with the clinical outcomes of COVID-19”, the authors have made a substantial comprehended contribution, which fil the gap knowledge. Each section is well discussed. The artwork is quite attractive.
Please find some of the suggestions that would improve the manuscript.
Minor comments
Carbohydrate recognition via DC-SIGN as PRR, this reference would be appropriate. https://doi.org/10.1016/j.intimp.2019.105684
Figure 4 legend is missing, please take care.
Page 10, TLR7… So probably TLR7 also be a selective target in SARS-CoV-2 infection. Perhaps this is the reason for “COVAXIN” [Bharat Biotech vaccine] efficacy. COVAXIN is composed of TLR7 agonist. Please include some discussion about this. https://doi.org/10.1016/S1473-3099(20)30942-7, https://doi.org/10.1016/S1473-3099(21)00070-0
Mycobacterium avium should be italics.
Table 1: “SARS-CoV1”------- “SARS-CoV-1”
“SARS-CoV2”------ “SARS-CoV-2”
Table 3: COVAXIN should be listed.
Author Response
1.The suggested reference was not included because several sources are already sited for discussion of DC-SIGN as PRR.
2.Figure 4 legend added.
3.The sources we found discussed COVAXIN as an inactivated vaccine but we did not find discussion on COVAXIN as a TLR7 agonist.
4.Mycobacterium avium italicized.
5.Table 1: “SARS-CoV1”------- “SARS-CoV-1” fixed
6.“SARS-CoV2”------ “SARS-CoV-2” fixed
7. Table 3: COVAXIN added.